



# Not All Types of Secondary Organic Aerosol Mix: Two Phases Observed When Mixing Different Secondary Organic Aerosol Types

Fabian Mahrt[1,2], Long Peng[1,3], Julia Zaks[1], Yuanzhou Huang[1,†], Paul E. Ohno[4,5], Natalie R. Smith[6], Florence K.A. Gregson[1], Yiming Qin[4,§], Celia L. Faiola[6], Scot T. Martin[4,7], Sergey A. Nizkorodov[6], Markus Ammann[2] and Allan K. Bertram[1,*]

[1]Department of Chemistry, University of British Columbia, 2036 Main Mall, Vancouver, BC, V6T1Z1 Canada
[2]Laboratory of Environmental Chemistry, Paul Scherrer Institute, 5232 Villigen, Switzerland
[3]Institute for Environmental and Climate Research, Jinan University, Guangzhou 511443, China
[4]John A. Paulson School of Engineering and Applied Sciences, Harvard University, Cambridge, MA 02138, USA
[5]Center for the Environment, Harvard University, Cambridge, MA 02138, USA
[6]Department of Chemistry, University of California, Irvine, Irvine, CA 92697, USA.
[7]Department of Earth and Planetary Sciences, Harvard University, Cambridge, MA 02138, USA
[†]Now at: Anton Paar Canada Inc., 4920 Place Olivia, H4R 2Z8 Saint Laurent, Canada
[§]Now at: Department of Chemistry, University of California, Irvine, CA 92697-2025, USA

Correspondence to: Allan K. Bertram (bertram@chem.ubc.ca)

**Abstract.** Secondary organic aerosol (SOA) constitutes a large fraction of atmospheric aerosol. To assess its impacts on climate and air pollution, knowledge of the number of phases in internal mixtures of different SOA types is required. Atmospheric models often assume that different SOA types form a single phase when mixed. Here, we present visual observations of the number of phases formed after mixing different anthropogenic and biogenic SOA types. Mixing SOA types generated in environmental chambers with oxygen-to-carbon (O/C) ratios between 0.34 to 1.05, we found six out of fifteen mixtures of two SOA types to result in two phase particles. We demonstrate that the number of phases depends on the difference in the average O/C ratio between the two SOA types ($\Delta$(O/C)). Using a threshold $\Delta$(O/C) of 0.47, we can predict the phase behavior of over 90% of our mixtures, with one- and two-phase particles predicted for $\Delta$(O/C) < 0.47 and $\Delta$(O/C) ≥ 0.47, respectively. This threshold $\Delta$O/C value provides a simple parameter to predict if mixtures of fresh and aged SOA form one- or two-phase particles in the atmosphere. In addition, we show that phase separated SOA particles form when mixtures of volatile organic compounds emitted from real trees are oxidized.

## 1 Introduction

Secondary organic aerosol (SOA) particles make up 30–70% of submicron atmospheric aerosols by mass (Hallquist et al., 2009). SOA particles form through condensed phase processes and by gas phase oxidation of precursors, such as volatile organic compounds (VOCs), followed by gas–particle partitioning of the lower-volatility reaction products (Hallquist et al., 2009; Ervens et al., 2011).





The importance of SOA in the environment is widely recognized. SOA influences atmospheric chemistry (Pöschl and Shiraiwa,
2015) and negatively affects human health (Baltensperger et al., 2008; Nault et al., 2020). In addition, SOA particles play a
key role for climate (Kanakidou et al., 2005; Shrivastava et al., 2017) either directly, through scattering of incoming shortwave
radiation, or indirectly, by acting as cloud nuclei (Wolf et al., 2020; Lambe et al., 2011).

Due to the variety of biogenic and anthropogenic precursors and oxidation pathways involved in SOA formation, atmospheric
SOA particles usually contain a multitude of different oxygenated species (Nozière et al., 2015; Goldstein and Galbally, 2007).
Owing to this complexity, SOA ranks among the least understood types of aerosols (Chung and Seinfeld, 2002; Hallquist et
al., 2009). As an important example, the number of phases formed when different SOA types become internally mixed within
individual particles remains poorly understood. Different types of SOA can be internally mixed within individual particles
(Fig. 1a) by a range of processes, including coagulation of particles of different SOA types, gas–particle partitioning, multi-
phase chemical reactions and cloud processing (Marcolli et al., 2004a; Riemer et al., 2019; Ervens et al., 2008). We refer to
these particles as SOA+SOA particles.

Knowledge of the number and types of phases in SOA+SOA particles is needed to represent SOA formation in atmospheric
models and to assess their role for air quality, health and climate (Fig. 1b-e). For example, predictions of gas–particle parti-
tioning, and ultimately SOA mass in atmospheric models, depends on the number and types of phases. SOA formation in
models is often represented as absorptive gas–particle partitioning of semi-volatile species (Donahue et al., 2006; Odum et al.,
1996; Pankow, 1994; Bowman et al., 1997). In addition, most modelling studies (Strader et al., 1999; Carlton et al., 2010)
simply assume the formation of a single condensed organic phase when SOA types from different sources mix. In the case of
a single phase, the activities of the organic molecules are lowered, in turn lowering their equilibrium partial pressures. Con-
versely, the equilibrium partial pressure is greater if the different SOA types form separate phases due to repulsive intermo-
lecular forces between immiscible molecules. A lower equilibrium partial pressure shifts the thermodynamic gas–particle
equilibrium to the condensed phase and enhances uptake of semi-volatile vapors (Fig. 1b). Such enhancement will be smaller
or absent in case of two-phase organic particles, depending on whether the different SOA types are partially or completely
immiscible. Thus, models that assume a single organic phase can overpredict particulate organic mass concentrations if SOA
mixtures form multiple phases.

The presence of multiple organic phases can further affect the ability of the SOA particles to act as cloud nuclei, by impacting
the surface composition and surface tension, as the particles grow and activate into liquid cloud droplets, (Fig. 1c, Huang et
al., 2021; Ovadnevaite et al., 2017) with implications for cloud microphysical and radiative properties, i.e., their direct climate
effect. In addition, in phase-separated SOA particles with a core-shell morphology, the shell can shield the core (phase) from
the surrounding gas-phase, and can act as a resistor for mass transfer, impacting heterogeneous chemistry. As an example, the
presence of a SOA shell phase of low water content could considerably limit the reactive uptake of $N_2O_5$ into a SOA core



phase of higher water content (Fig. 1d, Folkers et al., 2003). Based on the discussion above, it is imperative to consider the number of phases in SOA+SOA particles when predicting the impact of SOA on air pollution, health and climate (Fig. 1e).

Despite the influence on predictions of organic mass concentrations, cloud formation ability, and heterogeneous chemistry, our knowledge of the number of phases in internal SOA mixtures, remains uncertain. Several previous studies investigated the miscibility of a single-component oxidized organic, which was used as a proxy of atmospheric SOA, with SOA material

generated in environmental chambers (Song et al., 2011; Ye et al., 2016a; Gorkowski et al., 2020, 2017; Ye et al., 2018a; Gordon et al., 2016) (Table S1). While complete mixing was observed in some experiments, other experiments reported distinct phases, i.e., no mixing. All of these studies were limited to SOA material generated from α-pinene ozonolysis, while in the atmosphere SOA forms from different precursors. In addition, ambient SOA is more complex than the single-component proxies used in these studies (Nozière et al., 2015).

The miscibility of two SOA types that were both generated in environmental chambers has only been investigated in a handful of studies (Table S2). In some of these studies (Ye et al., 2016b, 2018b; Robinson et al., 2013) two different SOA populations were independently produced and subsequently brought together to mix. In these studies, some mixing was observed, at least at high relative humidities (RH) when diffusion limitations within the particles were minimal. Nevertheless, these studies were not able to distinguish between single-phase particles (complete miscibility) and two-phase particles (partial or complete mis-

cibility), meaning that the number of phases could not be conclusively constrained. Thus, there is a real need for experiments that directly determine the number of phases visually after mixing two SOA types. There is also a real need to link the properties of the SOA types with the number of SOA phases for predictive purposes.

Here we used microscopy to observe the number of phases in particles containing mixtures of different SOA types. We produced different SOA types within environmental chambers from oxidation of single precursors, including α-pinene, β-caryo-

phyllene, farnesene, valencene, catechol and toluene. In addition, we investigated the phase behavior of SOA particles generated from oxidation of real tree emissions. We used the elemental oxygen-to-carbon (O/C) ratio to link the SOA properties with the number of phases, and show that two-phase SOA+SOA particles can form if the difference in the O/C ratio between the SOA types is sufficiently large.

## 2 Results and Discussion

Before studying SOA mixtures, we determined the phase behavior of the unmixed SOA types between 90% to 0% RH. The individual SOA types formed single-phase particles in this RH range (Fig. S5), and were present in an amorphous phase state, since the particles did not have irregular features that would indicate crystalline material. This is expected since an amorphous state is most often the thermodynamic stable phase in multicomponent organics (Marcolli et al., 2004b).


Depicted in Fig. 2 are example fluorescence microscopy images of internally mixed SOA+SOA particles recorded between
90% to 0% RH. The absence or presence of multiple phases was determined based on the absence or presence of multiple
colors within individual particles (Section A2.1). We observed particles with a single phase when SOA from α-pinene ozonol-
ysis was mixed with SOA from β-caryophyllene ozonolysis. By contrast, two phases formed when SOA from β-caryophyllene
ozonolysis was mixed with SOA from toluene photooxidation.

Figures S6–S10 show images of all 15 SOA+SOA mixtures studied. For all mixtures, there was no dependence of the phase
behavior on humidity between 90% to 0% RH. A summary is presented in Fig. 3. In 9 out of 15 mixtures we observed single-
phase particles, i.e., complete miscibility. In 6 mixtures we observed two-phase particles. In these cases, we cannot distinguish
if phase separation resulted in partial- or complete-immiscible phases.

To determine if multiple organic phases also form in more complex SOA, we performed additional phase behavior experiments
using SOA particles generated from photooxidation of VOCs emitted by real pine trees, as detailed in Smith et al. (in prep.).
Tree emissions involve complex blends of volatiles, which upon oxidation result in mixtures of different SOA types, thus
representing increasingly realistic SOA mixtures. Images from these experiments demonstrate that two SOA phases are present
in these particles for 90–0% RH (Fig. S11). This is consistent with previous observations of multiple organic phases in SOA
particles generated by photooxidation of complex VOC mixtures (R. Smith et al., 2021; Song et al., 2019). The combined
results further emphasize the importance of multiple phases in atmospheric SOA.

In Fig. 3b we show the number of phases as a function of the average O/C ratios of the two SOA types, to link the phase
behavior with chemical properties. Phase separated SOA+SOA particles were observed if the average O/C ratio of one SOA
type was high, and the average O/C ratio of the other SOA type was low. To further elucidate this dependence, we calculated
the difference in the average O/C ratios between the two SOA types, i.e., $\Delta(O/C) = |O/C_{SOA1}-O/C_{SOA2}|$. Here, $O/C_{SOA1}$ and
$O/C_{SOA2}$ are the average O/C ratios of SOA1 and SOA 2, respectively. Figure 4 shows the number of phases within our
SOA1+SOA2 particles as a function of the $\Delta(O/C)$ value. One- and two-phase SOA+SOA particles are associated with low
and high $\Delta(O/C)$ values, respectively, as empirically described by:

$$Y = (1-2)/(1+\exp[(\Delta(O/C)-\Delta(O/C)_{threshold})/m]) + 2. \tag{1}$$

The numbers in the numerator of Eq. 1 denote the physical limits of phases within the SOA1+SOA2 particles, $m$ denotes the
slope, and $\Delta(O/C)_{threshold}$ denotes the threshold below and above which one- and two-phase SOA+SOA particles form. A value
of $\Delta(O/C)_{threshold} = 0.47$ describes 14 out of 15 mixtures studied.





The $\Delta$(O/C) value of the real tree SOA is unknown because the molecules making up the real tree SOA particles are formed from different VOCs and reaction pathways, each yielding molecules with a different O/C ratio. Nonetheless, we expect molecules with a wide range of O/C ratios in the real tree SOA. Thus, the observation of multiple phases in these particles is consistent with our results from the SOA+SOA mixtures, which showed two-phase particles when the $\Delta$(O/C) was large.

A trend of the SOA+SOA phase behavior depending on the $\Delta$(O/C) is expected, because the average O/C ratios are related to the polarities and hydrophilicities of the SOA types, that in turn influence the phase behavior (Zuend et al., 2010; Ye et al., 2018a; Gorkowski et al., 2020, 2019). While other SOA properties could further help describing the phase behavior (Section S9), the O/C ratio of organics is known to impact the phase behavior of inorganic-organic mixtures (e.g., Song et al., 2012; You et al., 2012, 2013) and organic-organic mixtures (e.g., Gorkowski et al., 2020; Song et al., 2020). If the $\Delta$(O/C) of a SOA+SOA mixture is low, the polarities of the molecules within the mixture are often similar to each other. Following the principle of "like dissolves like" the SOA types in such cases are often similar enough to mix completely. Instead, if the $\Delta$(O/C) of a SOA+SOA mixture is large, the polarities of the SOA types are often different from each other, causing them to more likely form different polarity phases or mix only partially.

For phase-separated SOA+SOA particles, partial miscibility and complete immiscibility cannot be resolved from our experi-
ments. For sufficient contrast in the average O/C ratios of the two SOA types, complete immiscibility is expected to prevail. Nonetheless, the O/C ratios used here denote the arithmetic means of the distributions of O/C ratios of the individual molecules making up either SOA type. If the two SOA types have overlapping O/C ratio distributions, partial miscibility between the two SOA types is expected to prevail. In this case the more oxidized organics partition into a high-polarity organic phase, predominantly formed by the SOA type with the higher average O/C ratio. By comparison, the less oxidized organics partition
into a medium-polarity organic phase that is predominantly formed by the SOA type with the lower average O/C ratio.

## 3 Implications for Internal Mixtures of Fresh and Aged SOA

In the atmosphere, different air masses often contain SOA with different degrees of oxidation, that can become mixed during atmospheric transport. For instance, if air masses containing aged SOA are transported over urban or rural environments, particles containing internal mixtures of fresh and aged SOA can form (Fig. 1a)by a range of process including coagulation,
gas–particle partitioning, multiphase chemical reactions and cloud processing (Marcolli et al., 2004a; Riemer et al., 2019; Ervens et al., 2008).. Modelling studies, typically assume that the fresh and aged SOA form a single condensed phase (e.g. Pankow, 1994). The observations herein challenge this approach, suggesting that two organic phases of different polarity will form when internally mixing fresh SOA and aged SOA if the difference in their average O/C ratios is larger or equal to 0.47. For these cases, atmospheric models that assume a single organic phase and absorptive gas–particle partitioning of semi-
volatile species might overpredict condensed-phase organic mass concentrations, with implications for air quality and climate projections (Fig. 1).





Two main subclasses of ambient SOA types have been identified from AMS measurements, namely, semivolatile oxygenated organic aerosol (SV-OOA) and low-volatility oxygenated organic aerosol (LV-OOA, Jimenez et al., 2009). SV-OOA and LV-OOA have average O/C ratios between approximately 0.32 to 0.83 and 0.68 to 1.32, respectively (Canagaratna et al., 2015). SV-OOA represents fresh, less-oxidized secondary organic material, whereas LV-OOA represents aged, more oxidized SOA (Jimenez et al., 2009). In the troposphere, SV-OOA and LV-OOA often co-exist. Table 2 lists the O/C ratios of SV-OOA and LV-OOA observed in several field studies, along with the $\Delta$(O/C) between these subclasses. Assuming our $\Delta$(O/C) threshold of 0.47 applies to these subclasses, two phases are predicted in 3 out of these 9 cases. Taken together, the overall pictures that emerges from our results and those reported previously is that it is imperative to consider the presence of multiple SOA phases when predicting SOA formation and its environmental impacts. Nonetheless, further experimental and modelling studies are needed to fully explore and quantify associated environmental implications and address the uncertainties remaining from our study.

One of the uncertainties when extrapolating our results to the atmosphere is that all our experiments were limited to supermicron particles. Previous studies have shown that phase separation can becoming volume-restricted for particles with diameters below 40 nm (Kucinski et al., 2019). Thus, we anticipate that our results apply to particles larger than 40 nm, but further measurements on this topic are needed. In addition, our experiments were limited to average SOA-to-SOA mixing ratios between approximately 0.06 and 1.08. The dependence of the number of phases on the mass mixing ratio of the SOA types warrants additional studies. Lastly, further experiments are also needed to establish a better understanding of the impacts of SOA viscosity on the number of phases in SOA+SOA particles.

## Appendix A: Methods

### A1 Methods of generating internally mixed SOA+SOA particles

Six different SOA types were investigated here. Each SOA type was generated from oxidation of a commercial precursor by either ozonolysis or photooxidation of the corresponding vapor in different environmental chambers. The chambers used for SOA generation in this study included an environmental chamber at the University of British Columbia (UBC-EC), an oxidation flow reactor available at UBC (UBC-OFR) and an oxidation flow reactor at Harvard University (HU-OFR), which are described in detail in Section S2. Here, we refer to mixtures of two distinct SOA types as SOA+SOA particles, to emphasize their nature of combining two different SOA types that were each generated from oxidation of single precursors in environmental chambers, and to further to distinguish theses experiments from the more complex SOA mixtures generated from real tree emissions. To generate the internally mixed SOA+SOA particles we used two different approaches, that are described in more detail below.

### A1.1 Consecutive generation and impaction





In the first approach, one SOA type was produced and collected onto hydrophobic glass slides by inertial impaction. Subsequently, another SOA type was produced, and impacted onto the same glass slides, where the particles of the first SOA type had already been collected onto. We refer to this approach as consecutive generation and impaction method. In total, the

consecutive generation and impaction method was used in 12 out of the 15 SOA mixtures studied (Table 1). For this method, the different SOA types were generated in either the UBC-EC, the UBC-OFR or the HU-OFR. In Section S2.1 we describe each experimental setup in more detail, along with the conditions used to generate the different SOA types for the consecutive generation and impaction experiments (Table S3).

**A1.2 Simultaneous generation and impaction**

In the second approach two different SOA types were simultaneously generated in the same environmental chamber by simultaneously oxidizing two precursor gases in the UBC-EC at the same time, and collecting the SOA material by inertial impaction. We refer to this method as simultaneous generation and impaction. The phase behavior of the SOA+SOA particles was independent of the method used to generate and collect the particles (Section S3). Therefore, we performed most of our experiments using the consecutive generation and impaction method, due to the greater flexibility, e.g., ability to combine

SOA types generated from two different oxidants. In Section S2.2 we describe the experimental setup for the simultaneous generation and impaction experiments in more detail, along with the conditions used to generate the different SOA types (Table S4).

**A2 Microscopy analysis of particle phase behavior**

The phase behavior of the particles containing SOA mixtures was investigated as a function of RH, using a combination of

fluorescence and optical microscopy techniques.

**A2.1 Fluorescence microscopy phase behavior experiments**

For phase behavior analysis using fluorescence microscopy, the hydrophobic glass slides with the SOA+SOA particles were collected from the impactor and solutions of trace amounts of Nile red (9-diethylamino-5H-benzo[α]phenoxazine-5-one; concentration ≤ 10 mg L$^{-1}$) in ethyl acetate (Sigma Aldrich, purity > 99%) were nebulized (Meinhard, model: TR-30-C0.5) onto

the SOA+SOA particles to make them fluorescent. High purity nitrogen (Linde, grade 5.0) was used as a carrier gas during nebulization of the ethyl acetate solutions. This nebulization of Nile red onto the slides, resulted in droplets on the hydrophobic glass slides containing the different SOA types, ethyl acetate and trace amounts of Nile red. Previous work has shown that SOA is largely soluble in ethyl acetate (Matsunaga and Ziemann, 2009). To allow the ethyl acetate to evaporate, the glass slides were placed into a fume hood for 1–3 min, resulting in Nile red infused SOA+SOA particles with typical diameters

between 40 μm to 120 μm, that were used for phase behavior analysis. The nebulization step followed by evaporation of ethyl


acetate resulted in coalescence of the SOA particles present on the glass slides, further enhancing the formation of internal mixtures and leading to an overall increase in SOA+SOA particle size.

For phase behavior analysis the slides with the Nile red-containing SOA+SOA particles were mounted into a flow cell (Bio-Surface Technologies Corp., model: FC 81) that was coupled to a fluorescence microscope (Olympus, model: XI70, U-MNIB filter cube with excitation wavelengths 470–490 nm and emission wavelength > 515 nm). Photons emitted from the Nile red infused SOA+SOA particles were detected with a color charge-coupled device camera (Olympus, model DP80). Nile red is a solvatochromic dye, that fluoresces at different wavelengths, when embedded in chemically different aerosol phases(Greenspan et al., 1985; Teo et al., 2021). The fluorescence color (emission wavelengths) of Nile red preliminary depends on the polarity of the aerosol phase (Huang et al., 2021; Greenspan et al., 1985), but also on other factors, such as the pH value of the organic phase (Hekmat et al., 2008) and aggregation of the solvatochromic dye (Kurniasih et al., 2015; Ray et al., 2019). In our analysis, we constrained ourselves to use the presence of multiple distinct colors to indicate the presence of multiple phases in the internally mixed SOA+SOA particles, following our previous approach (Huang et al., 2021; Ohno et al., 2021; Mahrt et al., 2021, 2022). During the phase behavior experiments, a flow (~1.3 L min$^{-1}$) of nitrogen gas (Linde, grade 5.0) was continuously passed over the particles. The RH in of the nitrogen gas was controlled by first passing it through a bubbler system mounted in a temperature-controlled refrigerated bath (Thermo Fisher, model RTE-140) upstream of the flow cell. The temperature of the flow cell was equilibrated to room temperature (293 ± 2 K) and continuously measured by a thermocouple mounted to the flow cell (T-type, Omega, model FF-T-20-100). The RH that the SOA+SOA particles were exposed to was calculated from the temperature of the flow cell and the dewpoint temperature of the nitrogen gas, measured in-line using a chilled mirror hygrometer (General Eastern, model: M4/E4) mounted ~10 cm downstream of the flow cell.

**A2.2 Optical microscopy phase behavior experiments**

For phase behavior analysis using optical microscopy, a similar approach was used. In these cases, the glass slides with the SOA+SOA particles were taken from the impactor and placed in a custom-built flow cell, which was coupled to an optical microscope (Zeiss, model: Axiotech). The stainless-steel body making up the base of the flow cell, where the hydrophobic glass slide containing the SOA+SOA particles is mounted onto, was held at a constant temperature of 293 K for all our optical microscopy experiments, as measured with a thermocouple (T-type, Omega, model FF-T-20-100) attached to the flow cell. The temperature was held constant by circulating a coolant through the stainless-steel body, using a refrigerated bath (ThermoScientific, model: RTE-740). A stream (~1.2 L min$^{-1}$) of nitrogen gas (Linde, grade 5.0) was continuously passed over the particles. The RH of the nitrogen stream was controlled by passing the flow through a bubbler system, mounted in a refrigerated bath (ThermoScientific, model: RTE-140), and the RH was changed by adjusting the temperature of the coolant. The dew point temperature of the nitrogen stream exiting the cell was measured in-line, ~10 cm downstream of the flow cell, using a humidity sensor (Vaisala, model: HMT330). The RH within the particles was calculated from the dew point temperature of the nitrogen stream and the temperature of the flow cell. Phase behavior analysis by optical microscopy was used for the





experiments of the real tree samples. In addition, the optical microscopy setup was used to cross-check results of selected fluorescence microscopy phase behavior experiments.

At the start of all our phase behavior experiments, the particles were allowed to equilibrate at 90% RH for a period of ~3 min. The RH was then decreased, at a rate of around 0.5% RH min$^{-1}$. During the RH-ramps, images were captured with a digital camera coupled to the microscope at values of 90%, 70%, 50% 30% and 0% RH. In both the fluorescence and optical micros-copy setups, the humidity accuracy was ±2.5% RH, determined from measuring the deliquescence RH of pure ammonium sulfate particles (Sigma Aldrich, purity > 99%), and comparing against deliquescence RH values reported in the literature

(Cziczo and Abbatt, 1999; Wise et al., 2003).

**A3 Mixing ratios of SOA types within internally mixed SOA+SOA particles**

The mixing ratio of the different SOA types within internally mixed SOA+SOA particles can depend on the method used to generate the SOA mixtures, and can impact the phase behavior. Here, we used two different approaches to constrain the SOA-to-SOA mixing ratio for experiments performed using the simultaneous generation and impaction method and experiments

using the consecutive generation and impaction method.

For experiments performed by the consecutive generation and impaction method, the hydrophobic glass slides contained par-ticles that were mixtures of both SOA types (internally mixed SOA+SOA particles), as well as particles that contained only one of the two SOA types (externally mixed particles). To constrain the mixing ratios of the two SOA types within the inter-nally mixed SOA+SOA particles, we estimated the SOA-to-SOA volume ratios for individual phase-separated particles. This

was achieved from three-dimensional particle profiles derived from confocal microscopy analysis of phase-separated SOA+SOA particles, following our previous approach (Mahrt et al., 2022). Details are given in Section S4.1. Based on this analysis, the overall average SOA-to-SOA volume ratio for consecutive generation and impaction was approximately 0.39, with the average value ranging from 0.07 to 0.56 for the different SOA+SOA particle types. Furthermore, the SOA-to-SOA volume ratio range from 0.01 to 0.96 in the individual SOA+SOA particles (Table S5).

For experiments performed by the simultaneous generation and impaction method, the two types of SOA either become inter-nally mixed during the generation step within the UBC-EC or become internally mixed within individual particles during the impaction step onto the hydrophobic glass slide. This allowed us to estimate the SOA-to-SOA mixing ratio of the deposited particles from the optical particle counter (OPC)-based mass concentrations measured when the two SOA types were generated separately within the UBC-EC but using the identical experimental conditions as for the simultaneous generation and impac-

tion experiments. Details are provided in Section S4.2. We estimated the average SOA-to-SOA volume ratios to range approx-imately from 1.08 to 0.78 for our experimental conditions (Table S6).

**A4 Particle measurements with a high-resolution time-of-flight aerosol mass spectrometer**



The elemental O/C ratios of the individual SOA types were determined using a high-resolution time-of-flight aerosol mass spectrometer (HR-ToF-AMS, Aerodyne Inc., Canagaratna et al., 2015; Aiken et al., 2008, 2007; DeCarlo et al., 2006; Jayne

et al., 2000), hereafter referred to as AMS. Within the AMS, SOA particles are impacted onto a heated (~600 °C) filament, causing the nonrefractory constituents of the particles to vaporize. Electron impact ionization (~70 eV) was used to ionize the resulting vapors, which were then extracted by a time-of-flight mass spectrometer. The AMS measured the bulk chemical composition by collecting mass spectra using the MS mode. To correct for air interferences, particle filtered (Whatman, 1851-047, grade QM-A) air from the chambers was periodically sampled with the AMS. To determine the average elemental O/C

ratios of the individual SOA types, the ion-optical V-mode (V-shaped ion flight paths, single ion reflection; resolution of ~2400 $m/\Delta m$) AMS data was used, and processed using standard techniques (Canagaratna et al., 2015; Aiken et al., 2008; Allan et al., 2003). Specifically, the O/C ratios were determined by analyzing the AMS mass spectra in Igor Pro 8 (Wavemetrics Inc.), using the PIKA v1.23 and SQUIRREL 1.631 (Aerodyne Inc.) software packages, and applying the improved ambient method (Canagaratna et al., 2015).

For the individual SOA types generated in the UBC-EC and UBC-OFR the individual SOA types were sampled directly from the environmental chambers using the AMS. For the toluene SOA material generated in the HU-OFR, the SOA was first collected onto hydrophobic glass slides at Harvard University. After shipping the samples to the University of British Columbia, the SOA was washed off from the glass slides, and the O/C ratio was determined by sampling the re-aerosolized solution with the same AMS instrument that was used to determine the O/C ratios of the SOA types generated directly at UBC. The

elemental ratios for the toluene SOA determined from the washed-off and atomized SOA material are within the range of values reported in the literature for SOA from toluene photooxidation (Song et al., 2016; Hildebrandt Ruiz et al., 2015). The wash-off method was validated using SOA generated from catechol ozonolysis in the UBC-EC as described in Section S5.

Shown in Table 1 are the O/C ratios of the individual SOA types used in our experiments as determined with the AMS (Table S3). The elemental ratios are in reasonable agreement with values reported previously, for similar experimental conditions

(Shilling et al., 2009; Kundu et al., 2017; Chen et al., 2012, 2011; Lambe et al., 2015).

*Data availability.* All data is documented and available in the Supporting Information.

Author Contributions. FM prepared all figures and wrote the original draft of the manuscript with contributions from AKB. LP, YH and FM conducted phase behavior experiments and analyzed data. LP performed confocal microscopy experiments and analyzed data. PEO and YQ prepared toluene SOA samples. NRS prepared real tree SOA samples. FKAG performed SOA

wash-off experiments. JZ sampled and analyzed AMS data. All authors discussed and interpreted data and contributed to revising the original manuscript draft. AKB, STM, SAN, CF and MA supervised the project.

*Competing interest.* The authors declare that they have no conflict of interest.



*Acknowledgements.* The authors gratefully acknowledge access to the biological services laboratory at the UBC and thank Elena Polishchuk for coordinating instrument usage. We thank Claudia Marcolli for helpful discussions. We thank Milan

Coschizza Benny Ng, Des Lovrity and Pritesh Padhiar for technical support and for machining parts of the set up. We further thank David Tonkin for technical support with the AMS, and Donna Sueper for support with AMS data analysis.

*Financial support.* This work is part of a project that has received funding from the European Union's Horizon 2020 research and innovation programme under the Marie Skłodowska-Curie grant agreement No. 890200 (FM). AKB, YH and FKAG acknowledge funding from Natural Sciences and Engineering Research Council of Canada (NSERC) through grant

RGPIN/04441-2016. LP acknowledges support from the National Natural Science Foundation of China (No. 42105102). PEO acknowledges support from the Harvard University Center for the Environment through the Environmental Fellows program. STM acknowledges funding from Environmental Chemical Sciences of the Division of Chemistry of the U.S. National Science Foundation (NSF, grant ECS-2003368). NRS and SAN acknowledge funding from the U.S. NSF (grant AGS-1853639).





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





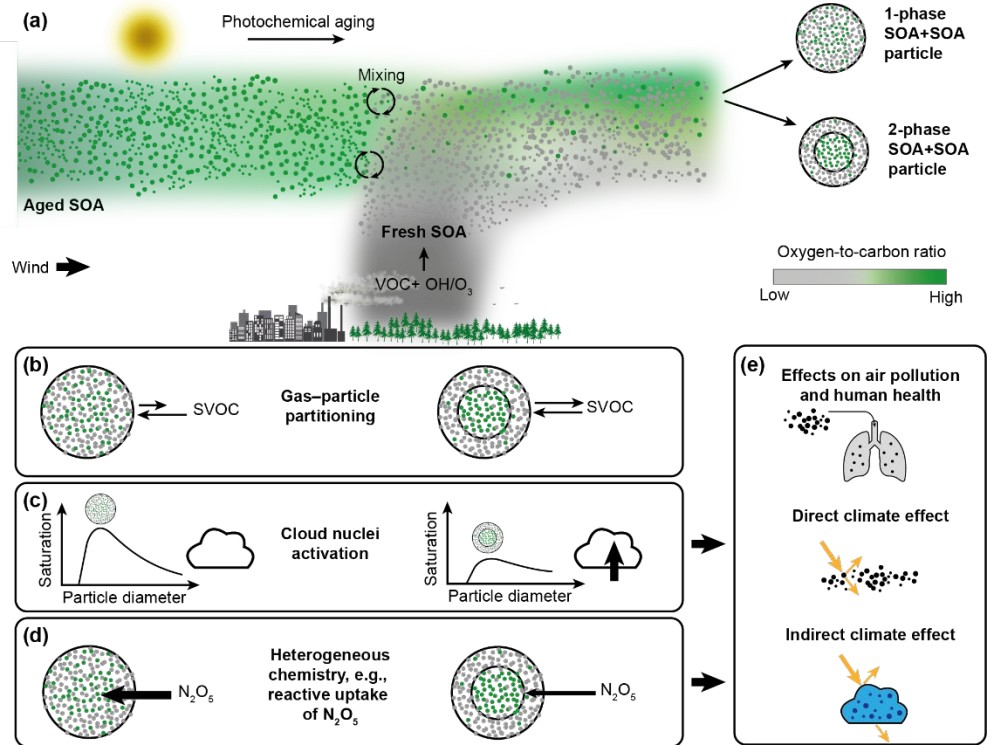


**Figure 1:** Formation of aerosol particles containing internal mixtures of different SOA types and impacts of the resulting number of condensed phases on air quality, human health and climate. (a) Mixing of air masses containing aged, highly oxidized background SOA material with air masses containing fresh, less oxidized SOA can result in SOA+SOA particles, with either one or two condensed phases, depending on the difference in average O/C ratio of the aged and fresh SOA types that become internally mixed. (b) Organic molecules in single-phase
SOA mixtures have reduced equilibrium partial pressures compared to when the two SOA types form distinct phases, allowing for an enhanced uptake of semi-volatile organic compounds (SVOCs). (c) The presence of a less oxidized organic phase making up the shell of a phase separated SOA particle can lower the surface tension and thus allow the phase-separated particles to activate into liquid cloud droplets at lower supersaturations compared to SOA particles with a single, homogeneous organic phase. (d) The presence of a less oxidized organic phase of low water content can act as diffusive barrier for mass transfer and affect heterogeneous chemistry, e.g., by limiting the reactive
uptake of $N_2O_5$. (e) Altogether, the presence of multiple phases in SOA particles can have important implications for the impact of SOA on air pollution, human health and climate. Figure based on Reid et al. (2018).

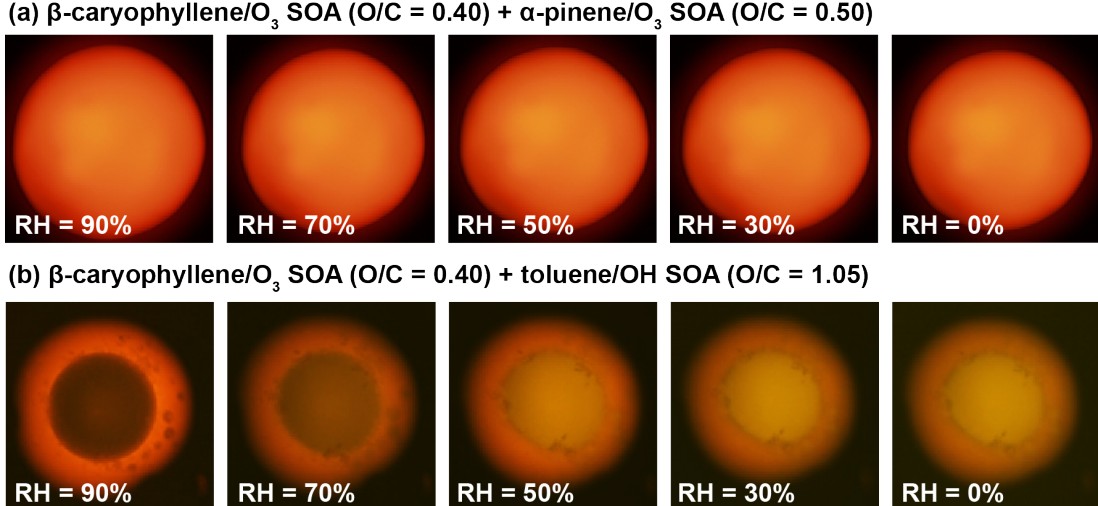

**Figure 2:** Example fluorescence microscopy images of internally mixed particles of two different types of secondary organic aerosol (SOA) material at different relative humidity (RH) levels. (a) Mixture of SOA materials derived from β-caryophyllene and α-pinene ozonolysis, showing single-phase particles between 90% to 0% RH. (b) Mixture of SOA material derived from β-caryophyllene and catechol ozonolysis, showing phase-separated particles between 90% to 0% RH. The colors result from trace amounts of Nile red within the SOA+SOA particles and the presence of different colors indicates the presence of multiple phases. Also indicated above each row is the average elemental oxygen-to-carbon (O/C) ratio of each SOA type, as determined from aerosol mass spectrometer measurements. Images were recorded for RH values decreasing from high to low values. The small inclusions in the outer phase of the β-caryophyllene SOA and toluene SOA mixture represents an emulsified, transient morphology.



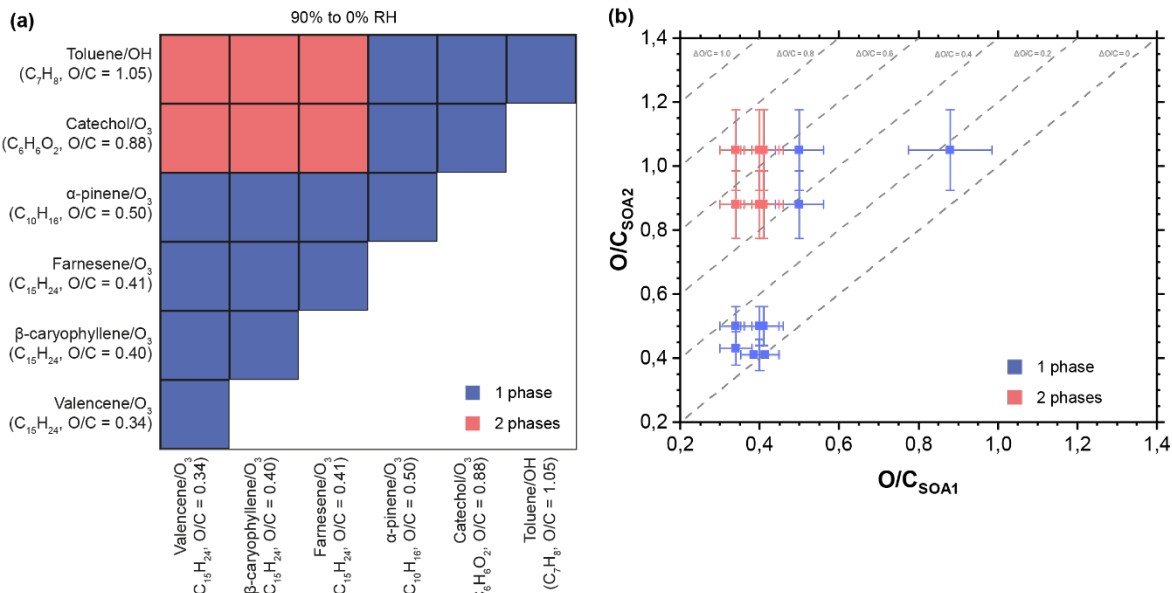

**Figure 3:** Summary of phase behavior of internal mixtures of two different types of secondary organic aerosol (SOA) materials. (a) Experimental matrix showing the number of (non-crystalline) phases for the SOA+SOA mixtures studied for relative humidities (RH) between 90% to 0%. Given along with each SOA type is the average elemental oxygen-to-carbon (O/C) ratio, the chemical formula of the precursor gas and the oxidant used to oxidize the precursor. (b) Number of phases observed for RH between 90% to 0% for the SOA+SOA particles as a function of the average O/C ratio of the two SOA types. Error bars indicate the 12% relative uncertainty of the average O/C ratio of each SOA type (average absolute value of relative error for multi-species organic aerosol)[55], as determined with the aerosol mass spectrometer (Table S3). The dotted lines indicate isolines of the differences in the average O/C of two SOA types within a mixture, $\Delta(O/C)$, and are given for intervals of $\Delta(O/C) = 0.2$. In all panels blue and rose symbols mark internal SOA+SOA mixtures with one and two phases, respectively. See Figs. S6–S10 for fluorescence images of all SOA+SOA mixtures.



**Figure 4:** Summary of the number of phases observed in our internally mixed SOA+SOA particles as a function of the absolute difference in the average O/C ratios between the two SOA types within a mixture, $\Delta(O/C) = |O/C_{SOA1}-O/C_{SOA2}|$. The horizontal error bars indicate the propagated error from the 12% relative uncertainty of the O/C ratios determined for each SOA type (average absolute value of relative error for multi-species organic aerosol; Table S3)[55]. The black, dashed line shows an empirical fit of the form of Eq. (1) to our phase behavior data, by choosing a steep slope (m = 4.85·10-5), so that the expression in Eq. (1) can be regarded as a step-function. The data points are randomly vertically offset around values of 1 and 2 for clarity and to reduce the overlap between the data points.

585





590  **Table 1**: List of phase behavior experiments of internally mixed particles containing two different types of secondary organic aerosol (SOA) material. Tabulated are the experiment number and the collection method, where "sim" denotes simultaneous generation and impaction, and "con" denotes consecutive generation and impaction. Also listed are the average O/C ratios for the individual SOA types, as measured when only one VOC was present within the chamber (Table S3). The associated uncertainty in the calculated average O/C ratios is 12% (average absolute value of relative error for multi-species organic aerosol)[55]. The values of Δ(O/C) denote the difference in the average O/C ratios of the two SOA types within a mixture.

| Exp. No. Collection method | SOA1 | SOA2 | $O/C_{SOA1}$ | $O/C_{SOA2}$ | $\Delta(O/C)$ | Number of phases between 90% to 0% RH |
|---|---|---|---|---|---|---|
| 1 sim | α-pinene/O₃ | β-caryo-phyllene/O₃ | 0.50 ± 0.06 | 0.40 ± 0.05 | 0.10 | 1 |
| 2 con | α-pinene/O₃ | Toluene/OH | 0.50 ± 0.06 | 1.05 ± 0.13 | 0.55 | 1 |
| 3 sim | α-pinene/O₃ | Catechol/O₃ | 0.50 ± 0.06 | 0.88 ± 0.11 | 0.38 | 1 |
| 4 sim | Catechol/O₃ | β-caryo-phyllene/O₃ | 0.88 ± 0.11 | 0.40 ± 0.05 | 0.48 | 2 |
| 5 con | Catechol/O₃ | Toluene/OH | 0.88 ± 0.11 | 1.05 ± 0.13 | 0.17 | 1 |
| 6 con | β-caryo-phyllene/O₃ | Toluene/OH | 0.40 ± 0.05 | 1.05 ± 0.13 | 0.65 | 2 |
| 7 con | β-caryo-phyllene/O₃ | Farnesene/O₃ | 0.40 ± 0.05 | 0.41 ± 0.05 | 0.01 | 1 |
| 8 con | Farnesene/O₃ | α-pinene/O₃ | 0.41 ± 0.05 | 0.50 ± 0.06 | 0.09 | 1 |
| 9 con | Farnesene/O₃ | Catechol/O₃ | 0.41 ± 0.05 | 0.88 ± 0.11 | 0.47 | 2 |
| 10 con | Farnesene/O₃ | Toluene/OH | 0.41 ± 0.05 | 1.05 ± 0.13 | 0.64 | 2 |
| 11 con | Valencene/O₃ | Farnesene/O₃ | 0.34 ± 0.04 | 0.41 ± 0.05 | 0.07 | 1 |
| 12 con | Valencene/O₃ | α-pinene/O₃ | 0.34 ± 0.04 | 0.50 ± 0.06 | 0.16 | 1 |
| 13 con | Valencene/O₃ | β-caryo-phyllene/O₃ | 0.34 ± 0.04 | 0.40 ± 0.05 | 0.06 | 1 |
| 14 con | Valencene/O₃ | Catechol/O₃ | 0.34 ± 0.04 | 0.88 ± 0.11 | 0.54 | 2 |
| 15 con | Valencene/O₃ | Toluene/OH | 0.34 ± 0.04 | 1.05 ± 0.13 | 0.71 | 2 |



595 **Table 2**: Overview of O/C ratios reported for SV-OOA and LV-OOA components when sampling ambient organic aerosol. Elemental O/C ratios were calculated using the improved ambient method, and data taken from Canagaratna et al. (2015). Also indicated is the $\Delta$(O/C) value between the SV-OOA and LV-OOA components and the number of phases predicted using an $\Delta$(O/C) threshold value of 0.47 (Fig. 3).

| O/C ratio of SV-OOA | O/C ratio of LV-OOA | $\Delta$(O/C) | Number of phases predicted | Reference |
|---|---|---|---|---|
| 0.83 | 1.32 | 0.49 | 2 | DeCarlo et al. (2010)(DeCarlo et al., 2010) |
| 0.32 | 0.86 | 0.54 | 2 | Docherty et al. (2011)(Docherty et al., 2011) |
| 0.46 | 0.68 | 0.22 | 1 | Gong et al. (2012)(Gong et al., 2012) |
| 0.60 | 0.76 | 0.16 | 1 | He et al. (2011)(He et al., 2011) |
| 0.49 | 0.80 | 0.31 | 1 | Huang et al. (2011)(Huang et al., 2011) |
| 0.45 | 0.81 | 0.36 | 1 | Huang et al. (2012)(Huang et al., 2012) |
| 0.41 | 0.98 | 0.57 | 2 | Mohr et al. (2012)(Mohr et al., 2012) |
| 0.51 | 0.78 | 0.27 | 1 | Sun et al. (2011)(Sun et al., 2011) |
| 0.56 | 0.65 | 0.09 | 1 | Wang et al. (2010)(Wang et al., 2010) |