# Peer review of "Not All Types of Secondary Organic Aerosol Mix: Two Phases"

_Atmospheric Chemistry and Physics, 2022_

## Author Comment (AC1)

Dear editor,

Thank you for the assessment of our manuscript for publication as an ACP letter. Below we provide detailed responses to address each reviewer's comment separately. We numbered the main reviewer comments for ease of cross-referencing, e.g., R1C1, refers to comment 1 of reviewer 1. Our responses are highlighted in blue, changes to the manuscript are shown in *italics*, and line numbers refer to the revised manuscript (version without track changes).
* * *
**Reviewer 1 comments:**

General:

The authors show that phase separation can occur when SOAs from different VOC precursors are mixed. The manuscript describes new and original data and set it well into context of previous work.

The manuscript left over some questions, however for answering that the data set is too small and too selected. Nevertheless, it a very well written and inspiring manuscript, which could induce more research in the direction of phase separation in atmospheric OA. The manuscript is suited to be published as ACP letter after the authors considered some comments.

We very much thank the reviewer for their time and overall constructive and positive comments, which greatly helped us to clarify important aspects of our work. Detailed response to the individual comments are given below.

I would like the authors to respond to the following considerations.

**R1C1:** Hypothesis: the difference in average O/C ratio is a necessary prerequisite but not sufficient for phase separation. The width of the O/C distribution of the condensed products, thus the (non-) overlap in O/C must play a role, too. They authors addressed that a bit in the section where they discuss partial miscibility. It could be extented there. (For the moment we can neglect that O/C is not the real driver -only the indicator- for specific intermolecular interactions between differently functionalized molecules, which determines the phase behavior besides entropic aspects.)

We agree with the reviewer that next to the average O/C ratio, the widths and (non-)overlap of the O/C ratio distributions of the individual molecules making up the two SOA types likely play a role for the phase behavior of SOA+SOA particles. To address the reviewer's comment, we have modified the text on L141-143 to read (L143):

*The O/C ratios used here denote the arithmetic means of the distributions of O/C ratios of the individual molecules making up either SOA type. For sufficient contrast in the average O/C ratios and non-overlapping O/C ratio distributions of the two SOA types, complete immiscibility is expected to prevail. By contrast, if the two SOA types have overlapping O/C ratio distributions, partial miscibility between the two SOA types is expected to prevail.*

To further emphasize the potential importance of the widths of the O/C ratio distributions of the two SOA types, we have also added the following statement (L149):

*Thus, in addition to the average O/C ratios of the SOA types, the widths of their O/C ratio distributions of the condensed products, as well as the overlap of the distributions likely play a role for the phase behavior of SOA+SOA particles. Providing such detailed information together with phase behavior data is an important topic for future studies.*

In addition, we have added (L184):

*Lastly, there is a need to better understand how the O/C distributions of the condensed products within the SOA types impact the phase behavior.*

In order to address the reviewer's comment that the O/C ratio is the indicator, yet not the primary driver of the SOA+SOA phase behavior, we have revised the statement in L127-132 to now read (L132):

*While information on the more molecular interactions of oxidized molecules that drive the phase behavior, as well as other SOA properties beyond the O/C ratio of the organics could further benefit describing the phase behavior (Section S9), the O/C ratio of organics is a reasonable indicator of the phase behavior of inorganic-organic mixtures (e.g., Song et al., 2012; You et al., 2012, 2013) and organic-organic mixtures (e.g., Gorkowski et al., 2020; Song et al., 2020).*

**R1C2:** I guess the data set is a bit lucky in this way. The aromatics are shorter in C-chain and lower in molecular weight (MW) thus need to reach a relative high O/C before condensing. In contrast SQT will condense already at low O/C, because they have already high molecular masses *per se*. So *a priori* SQT SOAs are in tendency less polar than aromatic SOAs. These are exactly the two classes that do not mix, while MT with intermediate MW and oxidation degrees will mix with both other classes.

Thank you for this comment. This is definitely a reasonable suggestion. To investigate and clarify this, additional studies would be needed. To address this comment, we have pointed out at the end of the manuscript that studies with additional SOA types would also be useful, by adding the following statement (L184):

*Future studies with additional SOA types would also be useful to increase the size of the data set used to develop the Δ(O/C) framework, and to confirm the Δ(O/C) framework applies to a wide range of molecular weights and functional groups of the condensed-phase products.*

**R1C3:** Since all pure systems show no phase separation, different O/C bins in these cases are obviously not populated enough. As consequence O/C variation of groups of molecules within the product spectrum is too narrow. A Delta O/C = 0.47 and cannot be realized in order to undergo phase separation.

So, it is probably not only the Delta O/C but also the spread of the O/C around the average which is eventually important. From this point of view, it would be interesting to see if mixing SQT, MT and Aromatic SOA would lead to phase separation or if the MT products would "mediate" sufficient miscibility.

These considerations do not affect the results as such, but possibly a simple transfer to atmospheric mixtures and particles.

Please see our response to R1C1 above regarding the width of the O/C distribution. In addition, we have pointed out at that studies with ternary mixtures of SOA types would also be interested, by adding (L184):

*Experiments with ternary mixtures of different SOA types would also be interesting.*

**Minor comments:**

Looking at all the images I don't recognize the growth with increasing RH (compare suppl. Line 265-270). Can it be that transport barriers arising by the fact that the particles are in glassy or high viscous states state help to establish phase separations?

The reviewer is correct that the change in particles size for our SOA particles for humidities between 90% to 0% RH was small. This can be due to multiple reasons. One possibility is that the change in size appears small due to a change in the contact angle between the SOA material and the substrate as the RH changes. To check this possibility, contact angle measurements on the hydrophobically coated glass substrates should be performed as a function of RH. This was not possible here, as RH control is currently not available in our confocal microscope setup that can be used for contact angle studies and that is described in Sect. S4.1. Another possible reason is that the hygroscopic growth of the SOA is low in the RH range between 90% to 0%. Another possible reason could be related to viscosity. Viscosity is not expected to limit the water partitioning at the highest RH used here since the viscosity of SOA is most often small at high RH values. On the other hand, at 0% RH and room temperature, viscosity could limit partitioning of water vapor. At the end of the revised manuscript, we have pointed out that additional studies are needed to understand the effect of viscosity on the number of phases in the particles (L180).

*Further experiments are also needed to establish a better understanding of the impacts of other SOA properties, such as viscosity or molecular weight on the number of phases in SOA+SOA particles.*

Line 65: Anttila et al, J. Atm. Chem., 2007 show direct experimental evidence of phase separation and shell core morphology.

Thank you for pointing us towards this reference. We have added the reference to our manuscript.

**Technical corrections:**

Line 106/Figure S11: The images of the pine SOA look different in shape compared to the binary mixtures. It seems to wet the carrier surface? What is the reason?

Good catch! Most of the SOA samples collected from oxidizing VOCs emitted from real trees were collected on siliconized glass slides. These substrates have lower contact angles, compared to the more hydrophobic glass slides coated with either fluoropel-800 (Cytonix) or with Trichloro(1H,1H,2H,2H-perfluorooctyl)silane (Sigma Aldrich, 97% purity) that were used to collect the SOA+SOA mixtures. The lower contact angle of the siliconized glass substrates causes the real pine tree SOA to spread more across the glass slides compared to the SOA+SOA mixtures.

To clarify this, we have added a statement to the caption of Fig. S11 that these particles were collected on siliconized glass slides. In addition, the pine tree SOA material may contain molecules that act as surfactants, which can cause more spreading on the substrate surface.

Line 117/eq 1: Why "1-2" instead of "-1" ?

Please see our answer to R2C1 below.

Line 118: m denotes the slope in which sense? That it must become <<1 to get a step like function? Please, move that info from the caption Figure 4 (also) to here.

Please see our answer to R2C1 below.

Figure 3/line 576: The labels of the isolines are too small and too weak. They are hard to see.

We have increased the font size of the labels.

Line 241: "The RH within the particles…"? I guess, you meant "the RH the particles were exposed to"?

Correct, thank you. Changed accordingly.

**In the Supplement**

Line 71: I guess UV radiation at 254 nm is not energetic enough to photolyse O2. Could it be that the pen-ray lamp had also a 185 nm contribution?

The reviewer is correct, the UV-lamp emits light with wavelengths between 184.9 nm to 546.1 nm. What we referred to in the original manuscript is the mode of the emission spectrum of our UV-lamp. To avoid confusion, we have changed this to indicate the range of wavelengths emitted by the instrument and used for $O_2$ photolysis in the revised manuscript.

Line 324: Please, give a reference for the "TROPOS atomizer".

The atomizer used here is an instrument that is home-built by the Leipniz-Institut für Troposphärenforschung (TROPOS) that is similar in design to the Model 3076 atomizer distributed by TSI Inc. As there is no primary reference describing the atomizer we have revised the manuscript to now read (LS328):

*Specifically, the aqueous solutions were nebulized using an atomizer built by the Leipniz-Institut für Troposphärenfoschung (TROPOS; home-built atomizer, similar to Model 3076, TSI Inc.), fed through a custom-built diffusion drier containing molecular sieve…*

**Reviewer 2 comments:**

Concise write up covering the analysis of experiments investigating the miscibility of SOA material. However, in some parts the writing was brief to a fault. A few sections from the Supplement should be included in the main text for clarity. In particular, those that describe how the SOA+SOA mixtures were prepared. As it is now, the manuscript is very awkward to read and fully understand as the reader must alternate between two documents.

Thank you for your careful assessment of our work. Please note that the main text was intentionally kept succinct, to follow the guidelines for the ACP letter format. We have revised the manuscript at various places to improve clarity, but kept the length of the main text within the guidelines of ACP letter as required (please see the editor's comment posted in the interactive discussion, which is copied at the end of this document for convenience).

Below we provide detailed responses to your specific comments and suggestions, hoping to alleviate the reviewer's concerns.

**R2C1:** Only major point:

Eq. 1: Aside from the typo pointed out by the other reviewer, what is Y? By looking at Fig. 4 I guess Y is the 'number of phases'? If that's correct then this is way over complicated because you can only have two values. Just use a step function that is either 1 or 2 depending on if Delta O/C is below or above the Delta O/C threshold. This also gets rid of the 'm denotes the slope' nonsense.

To address the reviewer comment, we have simplified the mathematical description of our phase behavior data. Specifically, we have deleted the original equation (1) and instead used a step function to describe the data, as suggested (L112):

$$Number\ of\ non-crystalline\ phases = \begin{cases} 2, & \Delta(O/C) \geq 0.47 \\ 1, & \Delta(O/C) < 0.47 \end{cases} \tag{1}$$

*Equation 1, with a threshold Δ(O/C) value of Δ(O/C)threshold = 0.47 describes 14 out of the 15 mixtures studied (Fig. 4).*

**Numerous typos, grammatical errors. Here are a few:**

Line 146: two periods at the end of a sentence. Also that comma after 'studies' is unnecessary.

Corrected.

Line 147: 'that approach' instead of 'this approach'

Corrected.

Line 164: 'can become' instead of 'can becoming'

Corrected.
* * *
**Editor comment**:

Just a quick note that this article has been submitted according to ACP's 'letter' format, so has a strict word limit. While I would encourage the authors to take on board the reviewers' comments to help make the article as readable and accessible as possible, I recognise that it may not be possible to explicitly follow the suggestions of simply moving sections of the supplement to the main article, without breaking the word limit.

Thank you for the clarification.